



# ConcentrationTracker: Landlab components for tracking material concentrations in sediment

Laurent O. Roberge[1], Nicole M. Gasparini[1], Benjamin Campforts[2], and Gregory E. Tucker[3,4,5]

[1]Department of Earth & Environmental Sciences, Tulane University, New Orleans, 70118, USA
5  [2]Department of Earth Sciences, Vrije Universiteit Amsterdam, Amsterdam, 1081 HV, Netherlands
[3]Department of Geological Sciences, University of Colorado at Boulder, Boulder, 80309, USA
[4]Cooperative Institute for Research in Environmental Sciences, University of Colorado at Boulder, Boulder, 80309, USA
[5]Community Surface Dynamics Modeling System Integration Facility, University of Colorado at Boulder, Boulder, 80303, USA

10  *Correspondence to*: Laurent O. Roberge (lroberge@tulane.edu)

**Abstract.** We present a set of new Landlab numerical model components that allow users to track sediment properties across a landscape grid. The components use a mass-balance approach to partition the mass concentration of each property based on sediment fluxes calculated by various Landlab flux components. The methods are generic, allowing the user to assign any sediment property that can be expressed as a mass, volume, or number concentration (for example, mass of magnetite, volume of quartz, number of zircons, number of radiogenic $^{10}$Be atoms, "equivalent dose" of luminescence). Several properties can be tracked at once, each with concentration tracked in both sediment and bedrock at every location on the grid. Two ConcentrationTracker components have been formulated; one for distributed, space- and time-varying hillslope regolith movement and another for transport in fluvial networks, allowing for interaction between sediment in the water column and on the channel bed. These components can be used individually to study a single process or coupled to study the interactions of multiple processes acting on a dynamic landscape. We present two examples that illustrate the diverse uses of the ConcentrationTracker components: colour banding in hillslope regolith and provenance tracking of fluvial sediments.

## 1 Introduction

Numerical landscape evolution models (LEMs) are commonly used to study the form and evolution of topography. LEMs typically compute the movement and storage of sediment across a terrain surface (e.g., FastScape: Braun and Willett, 2013; TTLEM: Campforts et al., 2017; Badlands: Salles, 2016; CHILD: Tucker et al., 2001; SIBERIA: Willgoose et al., 1991). However, while some models track grain size populations (e.g., CAESAR: Coulthard et al., 2002; CHILD: Tucker et al., 2001), few LEMs account for the storage, fate, and transport of other sediment properties, such as lithology, geochemistry, or isotopic concentration. Enabling models to make predictions about sediment tracers and other properties would enhance our ability to interpret data and test hypotheses. Such a capability would be useful, for example, in modeling the propagation of source-to-sink sedimentary signals or understanding the effects of transient landscape response to cosmogenic nuclide concentrations.





As sediment travels from its source to its sink, properties such as isotope concentrations can change, necessitating tools that not only simulate sediment mass balance but also track the evolving characteristics of sediment.

Recently, a great focus has been placed on tracking cosmogenic nuclides, resulting in the development of several LEMs with this capability (Carretier et al., 2023; Mudd, 2017; Petit et al., 2023; Reed et al., 2023; Xie et al., 2022). A brief review of these

models is presented in Section 1.1 of this paper. The same mass balance theory used to conserve cosmogenic nuclide concentrations can be applied more generally to conserve concentrations of any passive tracer of sediment and allow model users to simulate many other sediment properties (concentration of zircons or other minerals within sediment, heavy metal contamination, etc.). A LEM that provides the basic tracking functionality and allows the user to define the property being tracked could be applied to a wide range of landscape evolution studies.

Here, we present the *ConcentrationTrackers*, a set of Landlab components for tracking concentrations of user-defined sedimentary tracers in a gridded landscape evolution model that includes a surface layer of mobile regolith overlying bedrock. The *ConcentrationTracker* components are designed to work with other Landlab components that compute sediment fluxes, either as a 2D field of flux per unit width (as computed, for example, by the *DepthDependentDiffuser* component to represent soil creep) and/or as flux along a network of channel segments (as computed for example, by the *SpaceLargeScaleEroder*

component to represent fluvial transport). These geomorphic components provide sediment fluxes to the *ConcentrationTrackers*, which use mass balance to transfer the passive sedimentary tracer concentrations across the landscape in the mobile regolith layer. The material being tracked can be any user-defined passive property of sediment that can be cast as a mass concentration (e.g., mass of magnetite per volume of sediment), volume concentration (e.g., volume of quartz grains per total volume of sediment), or number concentration (e.g., atoms of $^{10}$Be, number of zircons per volume of sediment). The

*ConcentrationTrackers* take advantage of the Landlab library to fill a niche un-supported by other concentration tracking models: a unit-agnostic approach that allows the user to define the property being tracked. This set of components is meant to be simple and generic, allowing the user to choose what transport processes and property concentrations are being modeled. In this paper, we show some examples ranging from a simple 1-dimensional hillslope profile showing downslope diffusion of a tracer pulse to a 2-dimensional catchment with fluvial erosion, transport, and deposition of sediment from two different

lithologies.

## 1.1 Review of models

Repka et al. (1997) developed a 2-dimensional numerical model of a catchment subjected to hillslope and fluvial sediment transport processes. They tracked cosmogenic nuclide concentrations in moving grains to study pathway-dependent changes across the landscape, though they assumed an equilibrium landscape morphology in which there are no changes to topography.

This approach has been followed up by many others (Ben-Israel et al., 2022; Carretier et al., 2009, 2019; Carretier and Regard, 2011; Codilean et al., 2010), but cannot be used to simulate transient topography or responses to external forcing on a landscape.



Small et al. (1999) added this possibility by including conservation of cosmogenic nuclide concentrations in a 1-dimensional numerical hillslope profile model with soil production and hillslope sediment transport. The hillslope profile approach was

expanded upon over the next decade and a half, with a strong focus on cosmogenic nuclide concentrations (Anderson, 2015; Campforts et al., 2016; Ferrier and Kirchner, 2008; Heimsath, 2006).

Mudd (2017) modeled $^{10}$Be and $^{26}$Al concentrations in a 2-dimensional LEM that simulated hillslope sediment transport and detachment-limited fluvial incision on a gridded topographic surface. Unlike the 1-dimensional hillslope profile models, this approach does not track a mobile regolith layer nor resolve vertical concentration changes. This has been followed up by

several other 2-dimensional cosmogenic nuclide-tracking LEMs, all with different approaches and potential uses (Carretier et al., 2023; Petit et al., 2023; Reed et al., 2023). Petit et al. (2023) used the Badlands model (Salles, 2016) to explore $^{10}$Be transport in a source-to-sink system. Badlands simulates hillslope sediment transport and fluvial incision with a similar single-surface detachment-limited approach to that of Mudd (2017) and includes a submarine deposition component. The 2-dimensional LEM of Reed et al., (2023) conserves cosmogenic nuclide concentrations (e.g., $^{10}$Be, $^{26}$Al, $^{14}$C) in a mobile regolith

layer overlying bedrock and includes chemical weathering and explicit calculation of profiles in the regolith layer. The model uses a detachment-limited threshold stream-power incision approach for fluvial transport. The Cidre model (Carretier et al., 2016) uses a Lagrangian approach to track individual grains seeded across a landscape and transported within sediment fluxes. The fluxes are calculated using an erosion–deposition approach to solve for hillslope and fluvial processes. In 2023, the model was updated to include tracking of concentrations of several cosmogenic nuclides ($^{10}$Be, $^{26}$Al, $^{21}$Ne, $^{14}$C, and others) within the

individual grains as they travel across a landscape (Carretier et al., 2023).

The effects of episodic spalling and mass wasting on sedimentary tracer concentrations can be significant and have been studied in the context of $^{10}$Be in several ways. Lal (1991), Brown et al. (1995), Small et al. (1997), and Reinhardt et al. (2007) use 0-dimensional models simulating cosmogenic nuclide concentration response to periodic spalling or mass wasting events that uniformly remove a specific depth of material. Francis et al. (2020) extended the processes in the 0-dimensional approach to

include stochastic earthquake-triggered landslides and regolith storage.

Niemi et al. (2005) and Yanites et al. (2009) used 2-dimensional catchment plan-view approaches to model the effects of spatially discrete landslide events on cosmogenic nuclide concentrations exported from the catchment. In both cases, landslide frequency and area were derived from power-law frequency–magnitude relationships. Landslides were located randomly throughout the domain without consideration for slope or aspect. Niemi et al. (2005) assumed a detachment-limited system

with no ability to cause topographic change and used the catchment for spatial statistics. On the other hand, Yanites et al. (2009) used a landscape evolution approach in which landslides erode material and transport it to the fluvial system. They used a mixing model to simulate fluvial storage of landslide-derived sediment but avoided modeling landslide deposits spatially.

Xie et al. (2022) used the CAESAR-Lisflood model to track the movements of landslide-derived sediment as it mixes with background fluvial and hillslope sediments. CAESAR-Lisflood is a cellular automaton landscape evolution model that uses a

diffusion equation for hillslope creep and a two-dimensional hydrodynamic model and a choice of sediment transport equations to simulate fluvial morphology (Coulthard et al., 2002, 2013; Van De Wiel et al., 2007). Landslides occur as rules-based



emergent events in which locally over-steepened slopes iteratively adjust to a pre-defined threshold angle by moving material downslope (Coulthard et al., 2002). Tracking functions have been implemented to allow tracking of grain size fractions, of heavy metal contaminants (Coulthard and Macklin, 2003), and of provenance from pre-assigned source areas (Xie et al., 2022).

With the *ConcentrationTracker* components, we take a more generalized approach than those described above. These passive sedimentary tracer components can be used in a Landlab-derived LEM to track the mass concentration of any user-defined sediment property.

## 2 Model description

The *ConcentrationTracker* set of components are mass balance models that define and track spatially variable concentrations

of sediment properties as a numerical landscape evolves. The landscape evolution is determined by one or more geomorphic transport models that simulate sediment flux processes in Landlab. The sediment fluxes are then used by the *ConcentrationTracker* components to redistribute concentrations accordingly. Two *ConcentrationTracker* components couple with two different flux components, the *DepthDependentDiffuser* and the *SpaceLargeScaleEroder*, to enable tracking from sediment transport by hillslope and fluvial processes (Table 1). The components may be used independently of each other or

may be coupled with one or more existing or future *ConcentrationTracker* components.

In this section, we summarize the Landlab modeling toolkit, then describe each *ConcentrationTracker* component along with a brief description of its corresponding Landlab flux component.

**Table 1. Landlab surface process components and their companion ConcentrationTracker components.**

| Process | Flux components | ConcentrationTracker component |
|---|---|---|
| Hillslope weathering, transport | DepthDependentDiffuser DepthDependentTaylorDiffuser | ConcentrationTrackerForDiffusion |
| Fluvial erosion, transport, deposition | SpaceLargeScaleEroder | ConcentrationTrackerForSpace |


### 2.1 Landlab modeling toolkit

Landlab is an open-source Python environment for modeling planetary surface processes (Barnhart et al., 2020; Hobley et al., 2017). It provides the core elements required for any surface dynamics model: a gridding engine, control of boundary conditions, and a modular set of individual surface process components that can be easily combined into multi-process models.

The gridding engine allows the user to create a model grid, store spatial data on the grid, and handle boundary conditions. The model grid contains nodes (points that can be regularly or irregularly spaced), cells (polygons that surround the nodes), and links (directional connections between pairs of nodes), as well as their dual complements (called corners, faces, and polygons). Data can be stored on any of these elements, for example surface elevation on nodes or directional sediment flux on links.





Nodes can be set to either "boundary" nodes or "core" nodes. Boundary conditions are then easily handled by defining the

boundary nodes as open, fixed-gradient, or closed boundaries.

In Landlab, individual surface processes are modeled by individual components. Since they all act on the same grid and use the same set of basic functions for data storage and manipulation, they can easily be combined and interact with each other in multi-process models.

Landscape evolution components in Landlab, like other LEMs, typically treat gravitational ("hillslope") and fluvial sediment

transport processes in different ways. Hillslope processes are commonly represented by calculating the volume flux of sediment per unit width across a terrain surface. When a numerical solution is implemented on a two-dimensional grid, the usual approach is to compute a volumetric flux per width between each adjacent pair of grid nodes. On the other hand, fluvial transport is often (though not always) represented in terms of water and sediment flow along a quasi-1D network of channel segments. In this case, the usual approach is to compute, for each grid cell, a volumetric sediment outflow rate, which is then

used as a sediment inflow for one its neighboring grid cells. In practice, this difference in the representation of sediment flow for hillslope versus fluvial processes necessitates two different implementations for the *ConcentrationTracker*: one designed to work with hillslope-process components or other components that use a distributed flux-per-width approach, and one for fluvial process components that rely on an embedded "routing network" approach. Below, we describe the general mass balance approach used for all *ConcentrationTracker* components followed by specific descriptions of the two different

implementations.

## 2.2 General mass balance approach

The *ConcentrationTracker* components follow a common mass balance foundation but differ in their respective details of mass transfer. The general mass balance equation is as follows:

$$\frac{\partial m_{Xs}}{\partial t} = M_{Xs_{in}} - M_{Xs_{out}} + \Psi_{Xs} \,, \tag{1}$$

where $m$ is mass (units of mass, M), $t$ is time (units of time, T), $M_{in}$ and $M_{out}$ are, respectively, the rate of mass transfer into and out of a defined area (M/T), and $S$ is the rate of mass gain or loss from sources and sinks within that area (M/T). The subscripts $Xs$ is used here to designate the sediment property of interest ($X$) carried by sediment ($s$) to differentiate from other similar variables. For example, $m_{Xs}$ is the mass of the property of interest carried by sediment, while $m_s$ is the mass of the sediment itself. Other materials that will appear in the equations in this paper are bedrock (subscript $r$), sediment produced by

weathering (subscript $p$), water (subscript $w$), and sediment entrained in the water column (subscript $sw$). A list of variables is in Appendix A.

The $\Psi_X$ term is defined by the user to allow specialized source and sink functions (for example, radionuclide production and decay) that may be independent of the specific sediment transport processes.



In both concentration tracking models, $m_{Xs}$ is the product of the volume of sediment, $V_s$ (L$^3$), and the mass concentration of the property carried by the sediment, $C_{Xs}$ (M/L$^3$). The governing equation for each *ConcentrationTracker* component when accounting for porosity, $\varphi$ (unitless) becomes:

$$\frac{\partial C_{Xs}V_s}{\partial t} = -\left(\frac{1}{1-\varphi}\right)\nabla \cdot Q_s C_{Xs} \,, \tag{2}$$

where sediment flux, $Q_s$ (L$^3$/T) is calculated by the pre-existing Landlab sediment flux process components, which are all briefly described below in association with the respective *ConcentrationTracker* component. A complete derivation can be found in the supplemental material.

### 2.3 Concentration tracker for hillslope processes

### 2.3.1 Hillslope processes in Landlab

Here, we present two Landlab model components that simulate hillslope transport processes acting on a mobile regolith layer overlying bedrock: *DepthDependentDiffuser* and *DepthDependentTaylorDiffuser* (see depth-dependent creep laws in Barnhart et al., 2019). The former simulates hillslope sediment transport using a depth-dependent linear diffusion approach in the style of Johnstone and Hilley (2015). The latter uses a depth-dependent non-linear diffusion approach, combining the concepts of Ganti et al. (2012) and Johnstone and Hilley (2015). Both components are designed for use with a separate external code (which could be another component) that computes the rate of conversion of bedrock into mobile regolith (or 'soil'). Given a mobile regolith layer, both components calculate a downslope sediment volume flux per unit width of that regolith, $q_s$ (L$^2$/T). For both components, the soil production rate, $P_s$ (L/T), must be applied as an input. In this paper, we calculate $P_s$ using the *ExponentialWeatherer* component, which follows an exponential production function in the style of Ahnert (1976):

$$P_s = P_0 e^{-H_s/H_d} \,, \tag{3}$$

where $P_0$ (L/T) is the maximum production rate, $H_s$ (L) is the depth of the regolith layer, and $H_d$ (L) is a depth–decay constant. $P_s$ is multiplied by the timestep duration to calculate a height of regolith produced over that time, which is added to the mobile regolith layer. Then, the sediment fluxes are calculated. For the *DepthDependentDiffuser*, the regolith transport rate is given by is:

$$Q_s = -DSH^*\left(1 - e^{-H_s/H^*}\right) \,, \tag{4}$$

where $D$ (L$^2$/T) is diffusivity, $S$ (L/L) is local slope, and $H^*$ (L) is regolith transport decay depth. The *DepthDependentTaylorDiffuser* replaces the above linear approach ($-DSH^*$) with a non-linear approach ($\frac{-DSH^*}{1-(S/S_c)^2}$) approximated using a multi-term Taylor series expansion:

$$Q_s = -DSH^*\left(1 + \left[\frac{S}{S_c}\right]^2 + \left[\frac{S}{S_c}\right]^4 + \cdots + \left[\frac{S}{S_c}\right]^{2(n-1)}\right)\left(1 - e^{-H_s/H^*}\right), \tag{5}$$





where $S_c$ (L/L) is the critical slope and $n$ is the user-defined number of terms.

Both hillslope diffusion components calculate fluxes on links between nodes. A $Q_s$ value at one link is both an outflux from the upslope cell and an influx to the downslope cell. Therefore, the two components generate the same three in/outfluxes: $P_s$
(an influx from the bedrock), $Q_s$ entering the cell from upslope (an influx), and $Q_s$ exiting the cell (an outflux). These are used in the *ConcentrationTrackerForDiffusion* mass balance described below.

**2.3.2 Mass balance**

Since the concentration is spatially variable and can be different between the bedrock and regolith layers, each of the in/outfluxes described above must have an associated concentration value. This weathered material associated with soil
production rate, $P_s$, acting on an area, $a$, has a concentration value $C_{Xp}$, that can be equal to the concentration in bedrock ($C_{Xr}$) or provided with a user-defined value or equation for scenarios in which the weathering process changes the concentration, such as chemical enrichment or depletion (Brimhall and Dietrich, 1987; Ferrier et al., 2011; Riebe et al., 2017). Each sediment flux is also associated with a concentration value, so the governing mass balance (Eq. 2) becomes:

$$\frac{\partial C_{Xs} V_s}{\partial t} = \frac{-\nabla \cdot Q_s C_{Xs} + P_s a C_{Xp}}{1 - \varphi}.$$ (6)

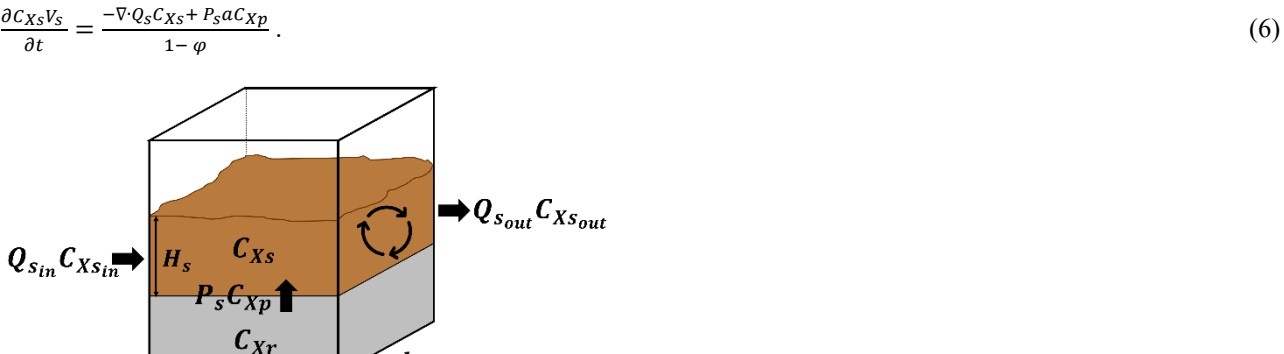


**Figure 1: Conceptual sketch of one cell with variables defined. Black arrows show mass fluxes that contribute to changes in concentration.**

**2.3.3 Numerical implementation**

Equation 6 is solved numerically using a first-order finite-volume approach that can act on most of Landlab's built-in grid
types (e.g., rectilinear, hexagonal). The following discretization shows the numerical approach applied to a simplified 1-dimensional example with spatial dimension $x$. With one less spatial dimension, sediment flux is now expressed as $q_s$ (L$^2$/T):

$$\frac{C_{Xs_i}^{t+1} H_{s_i}^{t+1} - C_{Xs_i}^t H_{s_i}^t}{\Delta t} = \frac{\left[ -\frac{q_{s_{i+1/2}}^{t+1} C_{Xs_{i+1/2}}^{t+1} - q_{s_{i-1/2}}^{t+1} C_{Xs_{i-1/2}}^{t+1}}{\Delta x} \right] + P_{s_i}^{t+1} C_{Xp_i}^{t+1}}{1 - \varphi},$$ (7)

where $t$ is the current timestep, $t+1$ is the next timestep, $i$ is the current node, and $i-1$ is the upslope node. Solving for $C_{Xs_i}^{t+1}$:





$$C_{Xs_i}^{t+1} = \frac{\Delta t}{H_{s_i}^{t+1}(1-\varphi)}\left[-\frac{q_{s_{i+1/2}}^{t+1}C_{Xs_{i+1/2}}^{t+1} - q_{s_{i-1/2}}^{t+1}C_{Xs_{i-1/2}}^{t+1}}{\Delta x}\right] + C_{Xs_i}^{t}\frac{H_{s_i}^{t}}{H_{s_i}^{t+1}} + \Delta t C_{Xp_i}^{t+1}\frac{P_{s_i}^{t+1}}{H_{s_i}^{t+1}(1-\varphi)}.$$ (8)

Since all flux, $q_s$, and height, $H_s$, values for $t+1$ are known (having been calculated by the *DepthDependentDiffuser* or *DepthDependentTaylorDiffuser*), the remaining unknown is $C_{Xs_i}^{t+1}$ on both sides of the equation. Using a first-order forward Euler method sets $C_{Xs_i}^{t+1}$ on the right-hand side equal $C_{Xs_i}^{t}$, the local concentration value at the current timestep, allowing us to solve for $C_{Xs_i}^{t+1}$ (on the left-hand side), the local concentration at the next timestep. This latter method requires us to assume

that the incoming sediment from upslope and from bedrock weathering fully mix with the local sediment already present before the resulting mix is fluxed onward to the next cell. This diffusive approach works for regolith-mantled hillslopes over long timescales (Hanks et al., 1984; Pierce and Colman, 1986).

### 2.4 Concentration tracker for fluvial processes

#### 2.4.1 Fluvial processes in Landlab

The concentration tracker for fluvial processes is designed to work with the *SpaceLargeScaleEroder*, as well as potential future components that use a similar mass-balance formulation. *SpaceLargeScaleEroder,* which is an update to the *Stream Power With Alluvium Conservation and Entrainment* (*SPACE*) component (Shobe et al., 2017), is a mass conservative erosion-deposition fluvial sediment transport model that acts on a mobile sediment layer and an underlying erodible bedrock layer. Bedrock erosion and sediment entrainment and deposition are explicitly calculated, allowing direct calculation of $Q_s$ and of

alluvial layer thickness, in which concentration $C_{Xs}$ is tracked by the *ConcentrationTrackerForSpace*.

Mass must be conserved both for sediment in the water column and for sediment and rock on the channel bed. For the channel bed, the rate of change in topographic surface elevation, $\eta$ (units of length, L), over time is the sum of changes to bedrock elevation, $R$ (L), and sediment layer thickness, $H_s$ (L):

$$\frac{\partial \eta}{\partial t} = \frac{\partial R}{\partial t} + \frac{\partial H_s}{\partial t}.$$ (9)

This can be expanded to include the processes driving those changes:

$$\frac{\partial \eta}{\partial t} = U - E_r + \left(\frac{D_{sw} - E_s}{1-\varphi}\right),$$ (10)

where $U$ (L T$^{-1}$) is bedrock uplift rate relative to a given baselevel, $E_r$ is the erosion rate of bedrock, $E_s$ is the entrainment rate of sediment from the bed into the water column, $D_{sw}$ is the deposition rate of sediment from the water column (all L T$^{-1}$), and $\varphi$ (-) is sediment porosity.

Fluvial erosion of bedrock, $E_r$ (L T$^{-1}$), and sediment, $E_s$ (L T$^{-1}$), follow a unit stream power formulation modified by an erosional efficiency term that modulates the relative effectiveness of each process. As sediment thickness increases, covering more of the bedrock bed, erosion of that sediment asymptotically approaches a maximum entrainment rate, while the erosion



rate of the underlying bedrock declines toward zero. Fluvial sediment deposition rate, $D_{sw}$ (L T$^{-1}$), uses a volumetric sediment-to-water flux ratio and a net effective settling velocity parameter, $V$ (L T$^{-1}$), which accounts for turbulence and determines

sediment transport distance, following Davy and Lague (2009). A complete description of the component's mathematics is provided in Shobe et al. (2017).

Conservation of mass in the water column is as follows:

$$\frac{\partial (Q_{sw}/Q_w) h_w}{\partial t} = E_s + \left(1 - F_f\right)E_r - D_{sw} - \frac{\partial (Q_{sw}/B)}{\partial l} \ . \tag{11}$$

Here, $Q_{sw}/Q_w$ is the concentration of sediment in a water column of height $h_w$. We write this concentration as a ratio of

sediment flux to water flux to differentiate it from the concentrations in the *ConcentrationTrackers*. $F_f$ is a unitless fraction of fine sediment eroded from bedrock that becomes permanently suspended in the water column. $B$ is channel width, and $l$ is the streamwise spatial dimension, so $\partial/\partial l$ is the spatial derivative with respect to streamwise distance. An assumption is made that $\frac{\partial (Q_s/Q_w) h_w}{\partial t} = 0$ over the timescales of interest for landscape evolution models, so that the spatial gradient in sediment flux can be calculated as:

$$\frac{\partial (Q_{sw}/B)}{\partial l} = E_s + \left(1 - F_f\right)E_r - D_{sw} \ . \tag{12}$$

As with the diffusion equations, the sediment flux is necessary for tracking concentrations as sediment moves across the landscape (in this case downstream). The numerical implementation solves this equation moving downstream from top to bottom. Since sediment influx to any one node is equal to the sediment outflux of the upstream node, a local analytical solution can be implemented numerically at each cell of area $a$ (units of L$^2$):

$$Q_{sw_{out}} = \frac{\sum Q_{sw_{in}} + E_s a + E_r a}{1 + \frac{Va}{Q_w}} \ . \tag{13}$$

Figure 2 shows a diagram of one cell.





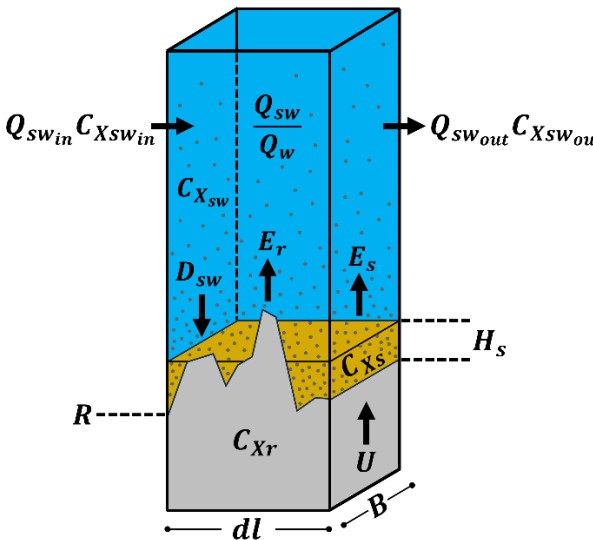

**Figure 2: Conceptual sketch of one cell with variables defined. Black arrows show mass fluxes ($D_{sw}$, $E_r$, and $E_s$) that transport concentrations ($C_{Xsw}$, $C_{Xr}$, and $C_{Xs}$, respectively) between parts of the cell and thus contribute to changes in concentration in sediment on the channel bed ($C_{Xs}$), in the water column ($C_{Xsw}$), and transported out of the cell by water flux ($C_{Xsw_{out}}$). Adapted from Shobe et al. (2017).**

### 2.4.2 Mass balance

Concentration is tracked in the layer of mobile bed sediment. The mass balance is directly affected only by sediment deposition from the water column ($D_{sw}$) and entrainment from the bed ($E_s$). Erosion of bedrock ($E_r$) does not directly impact the mobile bed layer, as it is first entrained into the water column. Therefore Eq. 2 becomes:

$$\frac{\partial C_{Xs} H_s}{\partial t} = \frac{C_{Xsw} D_{sw} - C_{Xs} E_s}{1 - \varphi} . \tag{14}$$

However, the concentration associated with sediment in the water column, $C_{Xsw}$, is unknown. This is calculated by applying a concentration mass balance to Eq. 11 for sediment conservation in the water column. We then use the same assumption that temporal change in mass is negligible when considering landscape evolutionary timescales and calculate the deposition term as $D_{sw} = \frac{Q_{sw}}{Q_w} V$, where V is a net effective sediment settling velocity parameter. This assumes that the speed of sediment and water are equal, so any changes in the $\frac{Q_{sw}}{Q_w}$ ratio must be driven by erosion and deposition. The result is a local solution to property concentrations associated with sediment suspended in the water column:

$$Q_{sw_{out}} C_{Xsw_{out}} = \frac{\sum Q_{sw_{in}} C_{Xsw_{in}} + E_s a C_{Xs} + (1 - F_f) E_r a C_{Xr}}{1 + \frac{Va}{Q_w}} . \tag{15}$$



This is the same local analytical solution as Eq. 13 for sediment fluxes, but now also tracks local concentrations of the user-defined sediment properties. The concentration in the water column ($C_{Xsw}$) is the same as that leaving the water column ($C_{Xsw_{out}}$), so $D_{sw}C_{Xsw}$ can now be applied to the bed concentration (Eq. 14), thus allowing us to solve for $C_{Xs}$.

### 2.4.3 Numerical implementation

We use a first-order finite-volume method to numerically solve Eq. 14 for most of Landlab's built-in grid types (e.g., rectilinear, hexagonal). For simplicity, we show the numerical discretization applied to a 1-dimensional example that assumes flow is from left to right:

$$\frac{C_{Xs_i}^{t+1}H_{s_i}^{t+1}-C_{Xs_i}^{t}H_{s_i}^{t}}{\Delta t}=\frac{C_{Xsw_i}^{t+1}D_{sw_i}^{t+1}-C_{Xs_i}^{t}E_{s_i}^{t+1}}{1-\varphi}, \tag{16}$$

where $t$ is the current timestep, $t+1$ is the next timestep, $i$ is the current location, and $i-1$ is the upstream location. Solving for $C_i^{t+1}$:

$$C_{Xs_i}^{t+1}=C_{Xs_i}^{t}\frac{H_{s_i}^{t}}{H_{s_i}^{t+1}}+\frac{\Delta t}{H_{s_i}^{t+1}}\left[\frac{C_{Xsw_i}^{t+1}D_{sw_i}^{t+1}-C_{Xs_i}^{t}E_{s_i}^{t+1}}{1-\varphi}\right]. \tag{17}$$

The value of $C_{Xsw_i}^{t+1}$ remains unknown, so a solution to Eq. 15 must be calculated. Here, $\sum Q_{sw_{in}}C_{Xsw_{in}}$ is known, as it is sum of outfluxes from upstream nodes (in this case, the outflux from the single upstream node at location $i-1$):

$$Q_{sw_i}^{t+1}C_{Xsw_i}^{t+1}=\frac{Q_{sw_{i-1}}^{t+1}C_{Xsw_{i-1}}^{t+1}+E_{s_i}^{t+1}aC_{Xs_i}^{t}+(1-F_f)E_{r_i}^{t+1}aC_{Xr_i}^{t+1}}{1+\frac{Va}{Q_w}}. \tag{18}$$

Solving for $C_{Xsw_i}^{t+1}$ provides the last piece of the puzzle to solve Eq. 17.

## 3 1-dimensional applications

Here we show 1-dimensional examples of the *ConcentrationTrackers* coupled with their respective companion flux components.

### 3.1 Hillslope processes

In this one-dimensional hillslope example, we couple the *ConcentrationTrackerForDiffusion* to the *DepthDependentDiffuser*. We generate a 200 m long hillslope that exists at a state of equilibrium with the local rock uplift rate using the parameters shown in Table 2. In this steady state, the rate of bedrock weathering is equal to the local rate of rock uplift relative to baselevel, such that the bedrock surface elevation remains steady in time. The increase in regolith depth caused by bedrock weathering is balanced by the rate of downslope regolith transport such that the regolith depth and regolith surface elevation also remain steady in time. Although the hillslope morphology is static in time, the sediment conveyer belt is constantly churning; bedrock



is constantly rising and weathering into mobile regolith, which is then transported downslope. The rock and sediment making
up the seemingly static hillslope are at no point static themselves. We show this effect with 3 examples of a 1-dimensional
hillslope profile (Table 2) and a packet of tracer sediment using the *ConcentrationTrackerForDiffusion*.

**Table 2. Parameters used for 1-dimensional hillslope example.**

| Parameter name | Symbol | Units | Value | | |
|---|---|---|---|---|---|
| Number of columns | $ncols$ | - | 20 | | |
| Spatial resolution | $dx$ | m | 10 | | |
| Temporal resolution | $dt$ | y | 1 | | |
| Uplift rate | $U$ | m/y | 0.00001 | | |
| Depth–decay constant | $H_d$ | m | 1 | | |
| Soil transport decay depth | $H^*$ | m | 1 | | |
| Maximum soil production rate | $P_0$ | m/y | 0.0001 | 0.0001 | 0.001 |
| Diffusivity constant | $D$ | m²/y | 0.01 | 0.005 | 0.005 |
| | | | **Ex. 1** | **Ex. 2** | **Ex. 3** |

In Example 1, the steady-state landscape comprises an approximately 20 m tall bedrock hillslope overlain by an approximately
2.3 m deep mobile regolith layer (Figure 3a). We place a virtual packet of tracer sediment at the 150 m mark by increasing the
concentration to a value of 1 (here representing a volume concentration) for the regolith layer. This can be thought of as digging
a virtual pit in the mobile regolith layer and replacing the removed material with tracer sediment, for example of a different
colour. Aside from colour, this sediment is exactly the same as that comprising the rest of the regolith layer. We then run the
numerical model for 10,000 years to track the downslope movement of this packet of tracer sediment through time (Figure
3d).

Since this example uses a linear diffusion equation, all transported sediment moves only from one node to the next downslope
before it can then be transported further. This results in a key assumption: the regolith layer is homogeneously mixed at all
times. There is no stratification of regolith and the process that causes downslope regolith movement of the soil also causes
full mixing of the regolith column. Although not presented as an example, this is true also of the non-linear diffusion model.
With homogeneous mixing, the tracer sediment becomes diluted as it travels downslope. With each increment of downslope
movement, any tracer sediment transported from upslope fully mixes with the local regolith layer before it can then be
transported further.

Since this is a steady-state hillslope, the rate of regolith production from bedrock weathering matches the rock uplift rate. This
means that the diffusivity constant, $D$, and the soil transport decay depth, $H^*$, both affect the steady-state topography of the
hillslope in order to equilibrate regolith flux rates. This is shown by comparing Example 1 (Figure 3a) with Example 2 (Figure
3b), where the hillslope is taller and steeper  in order to compensate for a smaller value of $D$ (Table 2; Figure 3b). Despite the
topographic change, the rate of movement of the sediment tracer pulse is unaffected by this change to $D$ (Figure 3d and Figure
3e) because the regolith layer depth and flux rates do not change. Example 3 (Table 2) shows a scenario (Figure 3c) in which
the maximum soil production rate, $P_0$, has increased by an order of magnitude. This increases the steady-state regolith depth



to 4.6 m. The tracer sediment pulse travels more slowly downslope, when compared to the thinner soils in Examples 1 and 2 (Figure 3a,b), as it gets diluted into a larger reservoir of non-tracer sediment at each incremental downslope movement (Figure 3f). Figure 3g shows a time series of tracer concentration as it exits the domain at the toe of the slope throughout the 10,000-year model run for each of the three examples. For Examples 1 and 2, concentration begins to increase after 1,000 years or so, when sufficient tracer sediment has made its way downslope to the toe. The tracer sediment pulse increases to a maximum near 3,000 years and then decreases again until about 6,000 years before tailing off toward zero again. In Example 3, the pulse moves slower, taking longer to start, to peak, and to return back to zero.

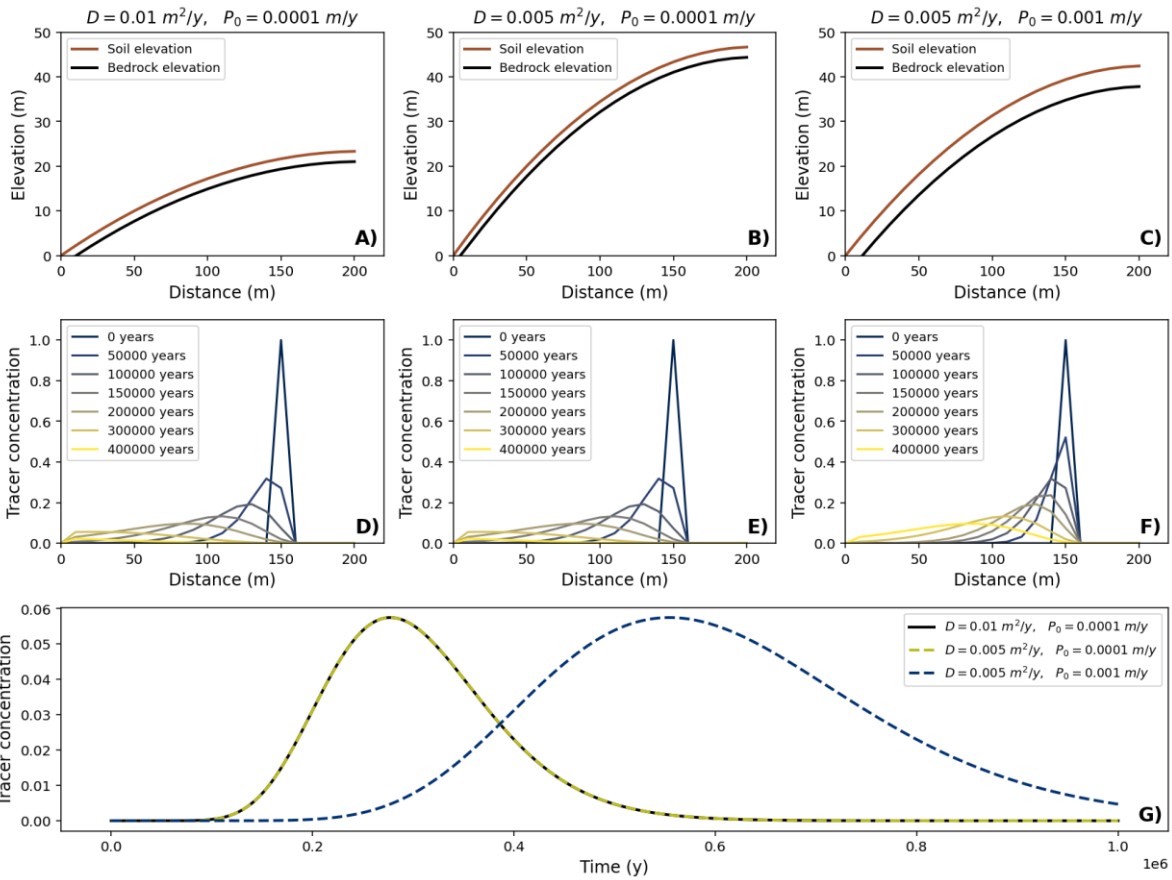

**Figure 3. Three examples illustrating the downslope movement of a tracer packet in a steady-state 1-dimensional hillslope profile. The top row (a, b, and c) shows the steady-state hillslope profile. The middle row (d, e, and f) shows the spatial location of the tracer packet through time as it travels downslope. The bottom row (g) is a time series comparing the concentration at the toe of the slope for the three examples through time. In Examples 1 and 2, despite differences in topography (a, b) the downslope movement of the tracer packet is the same (d, e), so the time series plot on top of each other in (g). In Example 3, the regolith layer is thicker (c), therefore causing the tracer packet to move more slowly (f) and move across the toe of the slope later (g).**



### 3.2 Fluvial processes

Here, we couple the *ConcentrationTrackerForSpace* to the *SpaceLargeScaleEroder* to produce a fluvial equivalent to the steady-state hillslope example described above. This time, we use the parameters in Table 3 to create a 2,000 m long river channel that exists at steady state. Bedrock erosion rate $E_r$ equilibrates to the local rate of rock uplift such that the bedrock surface elevation remains steady in time. Sediment generated from bedrock erosion is transported downstream and either deposited onto the channel bed (at a rate of $D_{sw}$) or exits the numerical domain through the outlet of the channel. Deposition of bedrock-derived sediment is balanced by erosion of channel bed sediment (at a rate of $E_s$) such that the thickness of the channel bed sediment layer remains constant. Although the bedrock and channel bed elevations remain unchanged through time, the bedrock is constantly being uplifted, eroded, and then transported downstream as sediment in the water column. The water column interacts with the channel bed by eroding and depositing sediment, so material is constantly moving throughout the system. Unlike the hillslope examples described earlier, the water column in *SpaceLargeScaleEroder* can transport sediment a long distance from its original location (i.e., more than the distance from one node to the next). This results in a sediment tracer pulse that acts differently than those in the hillslope examples. We show two examples below in which we place a packet of tracer sediment into the steady-state channel bed in a manner comparable to the hillslope examples (Figure 4).

**Table 3. Parameters used for 1-dimensional fluvial example.**

| Parameter name | Symbol | Units | Value | |
|---|---|---|---|---|
| Number of columns | $ncols$ | - | 20 | |
| Spatial resolution | $dx$ | m | 100 | |
| Temporal resolution | $dt$ | y | 1 | |
| Uplift rate | $U$ | m/y | 0.001 | |
| Sediment erodibility | $K_s$ | $m^{-1}$ * | 0.0002 | |
| Bedrock erodibility | $K_r$ | $m^{-1}$ * | 0.0001 | |
| Sediment porosity | $\varphi$ | - | 0 | |
| Fraction of fine material | $F_f$ | - | 0 | |
| Effective settling velocity | $V$ | m/y | 1 | 10 |
| * $m$ is the area scaling exponent for stream power. | | | **Ex. 1** | **Ex. 2** |

In Example 1 (Table 3), the steady-state river channel rises from 0 m to about 4 m over the course of its 2,000 m long path and is overlain by a bed sediment layer about 0.07 m thick (Figure 4a). We replace the bed sediment at the 1,500 m mark with a packet of tracer sediment by increasing the concentration to a value of 1. As with the hillslope examples, the concentration is unit agnostic but is imagined here as a volumetric colour concentration. In other words, the tracer sediment is identical to all other sediment in the model except for its colour, which is identified by a concentration value. We run the numerical model for 250 years to track the downstream movement of this packet of tracer sediment (Figure 4c). Some of the tracer sediment that is eroded from its original location is transported partway downstream before being deposited on the channel bed. This results in a small increase in concentration at each downstream node. However, unlike the hillslope example, some of the mobilized tracer sediment is transported far enough downstream that it leaves the numerical domain altogether in the first



timestep. This can be seen in Figure 4e, which shows the tracer pulse tracked at the outlet of the channel. The onset of the fluvial tracer pulse is immediate, and it peaks at 26 years. The pulse has largely decayed by 209 years, at which point only 0.01% of the original tracer remains in the channel bed.

The primary driver of the tracer sediment packet speed is the net effective settling velocity parameter ($V$), which controls the transport length scale for sediment entrained into the water column. Increasing $V$ causes sediment to travel a shorter distance before depositing, resulting in a tracer peak that takes longer to arrive at the outlet. In Example 2 (Table 3), we increase $V$ tenfold ($V = 10\ m/y$). Entrained sediment is very quickly redeposited, so much of the river's erosive capability is spent re-eroding bed sediment that has traveled only a short way downstream. In comparison to the first example, this creates a steady-state channel that is much steeper (reaching a maximum bedrock elevation of about 21 m) overlain by a bed sediment layer that is much thicker (about 0.24 m), shown in Figure 4b. The increased net effective settling velocity slows the tracer packet (Figure 4d) such that it takes 2 years for the first tracer sediment to reach the outlet (Figure 4e). The concentration at the outlet peaks at 61 years and decays back to 0.01% of the original tracer by 222 years (Figure 4e). At steady state, neither porosity of the channel bed layer, $\varphi$, (which affects the height of the bed sediment layer, but not its transport) nor the fraction of fine material, $F_f$, (which acts only on eroded bedrock material, not the bed sediment layer) have much effect on the tracer pulse.





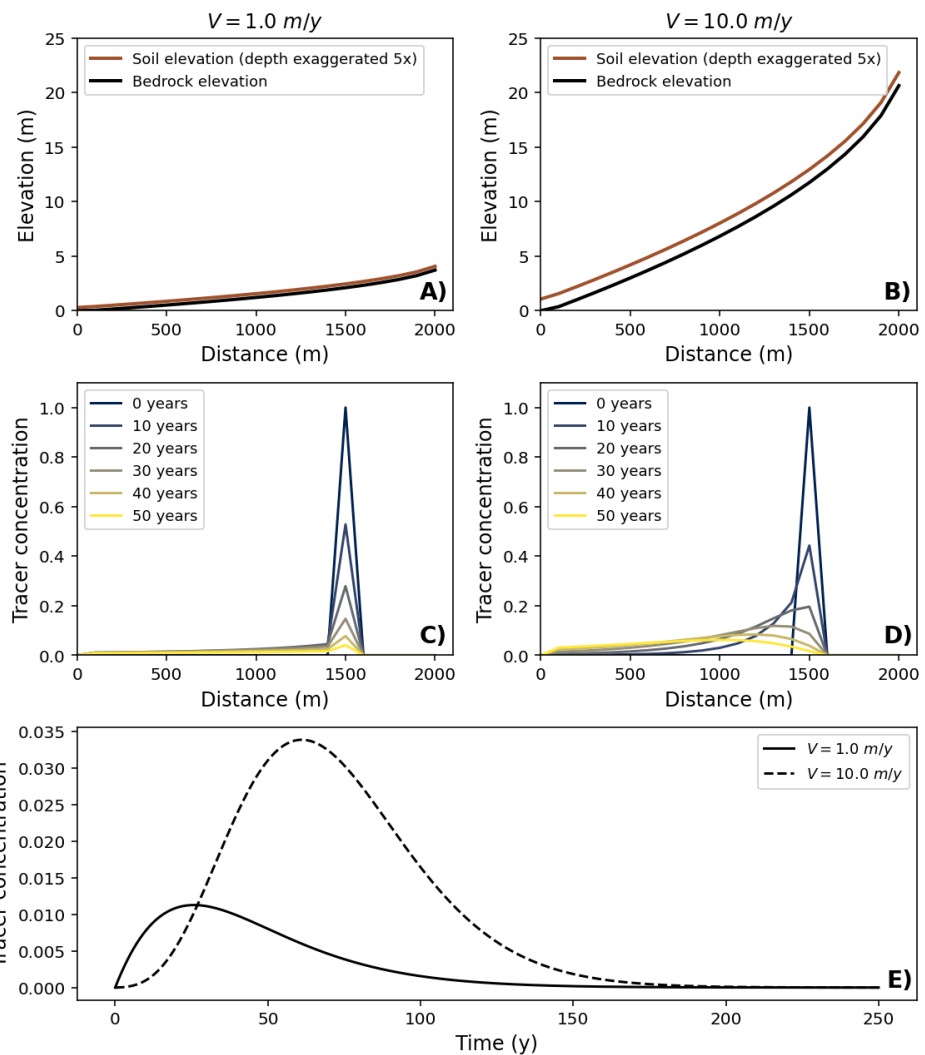


**Figure 4. Two examples illustrating the downstream movement of a tracer packet in a steady-state 1-dimensional stream channel profile. The top row (a and b) shows the steady-state channel profile with the depth of the bed sediment layer exaggerated by a factor of 5. The middle row (c and d) shows the spatial location of the tracer packet through time as it travels downstream. The bottom row (e) is a time series comparing the concentration at the outlet of the channel for the two examples through time. In Example 1,**
**the low value of the net effective settling velocity, $V$, causes most sediment eroded from the tracer packet to move far downstream with only a small fraction deposited along the way to the outlet. In Example 2, $V$ is increased tenfold, causing entrained sediment to become deposited not far downstream from its original location. The tracer packet therefore moves more slowly to the outlet.**

## 4 2-dimensional applications

Here we show 2-dimensional examples of the *ConcentrationTrackers*. For hillslope sediment transport processes, we illustrate
the effects of bedrock weathering on the surface expression of different coloured bedrock layers. For fluvial processes, we
show an example of bedrock provenance in which fluvial sediments are recruited from two regions of different bedrock colour.




We use colour as a simple visual tool. As explained before, the concentration values can be for any user-defined property of sediment that can be cast as a mass, volume, or number concentration.

## 4.1 Hillslope processes (hillslope colour bands)

In the 1-dimensional hillslope example, we placed tracer sediments into the mobile regolith layer to see their downslope transport. Here, we instead place the tracer 'colour' within the bedrock. We then allow the regolith to inherit colour from its parent bedrock through the weathering process, enabling us to see the surface expression of the bedrock colour.

To do this, we create an irregularly shaped hill on a 2-dimensional grid by setting a specific selection of grid nodes as open boundaries and evolving the landscape to a steady state over 200,000 years (hillshade shown in Figure 5a). We then apply two

bands of colour to the bedrock by changing the "*bedrock_property__concentration*" values from 0 to 1 at two specific elevation bands (Figure 5c). We use a yellow-to-red colourmap to roughly match the colours found in the Painted Hills of Oregon, USA (Figure 5b). All model parameters are shown in Table 4.

**Table 4. Parameters used for 2-dimensional hillslope example.**

| Parameter name | Symbol | Units | Value |
|---|---|---|---|
| Number of columns | $ncols$ | - | 41 |
| Number of rows | $nrows$ | - | 41 |
| Spatial resolution | $dx$ | m | 10 |
| Temporal resolution | $dt$ | y | 1 |
| Uplift rate | $U$ | m/y | 0.002 |
| Depth–decay constant | $H_d$ | m | 1 |
| Soil transport decay depth | $H^*$ | m | 1 |
| Maximum soil production rate | $P_0$ | m/y | 0.01 |
| Diffusivity constant | $D$ | m²/y | 0.5 |


We then evolve the landscape a further 10,000 years to see the colour of the surface sediment change as sediment is transported downslope and replaced by newly weathered bedrock from below. At the outset, there is a period of transient change in regolith colour as the landscape evolves from the initial condition to a new equilibrium state. Weathering of bedrock in place causes the concentration value to increase in the regolith overlying the two red bedrock layers as the newly produced regolith mixes

with the rest of the regolith column each timestep. Sediment transported downslope from above also mixes in, therefore increasing or reducing the concentration value at the downslope node depending on the upslope concentration. The result is a muted, diffuse-looking surface expression of the bedrock layers (Figure 5d). Immediately noticeable are the differences between the horizontally convex noses and horizontally concave "gullies". The regolith at a nose is mostly locally produced, as there is little to no supply from upslope. The concentration is therefore highly correlated with the underlying bedrock

concentration, resulting in very intense colours. On the other hand, the gully sediment is an integration of all the sediment transported from the surrounding upslope areas. This elevated level of mixing results in a smeared-looking surface expression of the underlying layers.





**Figure 5. Example of bedrock weathering and hillslope sediment transport on a 2-dimensional hillslope. (a) A hillshade of the irregularly shaped hillslope. (b) A picture of the Painted Hills in Oregon, USA. (c) An overlay on the hillshade showing the colour of the bedrock (the two red bands have a concentration value of 1, while the yellow regions have values of 0). (d) A hillshade overlay showing the steady-state regolith layer colour, which is the surface expression of the two red bedrock layers after weathering and diffusional hillslope sediment transport.**

## 4.2 Fluvial processes (provenance tracking)

Here, we again explore the surface expression of bedrock material, this time looking at fluvial channel bed sediments. We set up a 2-dimensional grid and close all boundary nodes except for one open outlet in the southwestern corner. The result is a river network that drains to this outlet (Figure 6a). We split the catchment into two regions: the northern third of the domain





has the "*bedrock_property__concentration*" value set to 1, indicating "red" bedrock, while the two thirds remaining to the south are left with a value of zero, indicating "yellow" bedrock (Figure 6b). Other than this colour difference, the bedrock

properties in the two regions are identical. All other model parameters are shown in Table 5.

**Table 5. Parameters used for 2-dimensional fluvial example.**

| Parameter name | Symbol | Units | Value |
|---|---|---|---|
| Number of columns | $ncols$ | - | 50 |
| Number of rows | $nrows$ | - | 50 |
| Spatial resolution | $dx$ | m | 100 |
| Temporal resolution | $dt$ | y | 1 |
| Uplift rate | $U$ | m/y | 0.001 |
| Sediment porosity | $\varphi$ | - | 0 |
| Fraction of fine material | $F_f$ | - | 0 |
| Effective settling velocity | $V$ | m/y | 1 |
| Area scaling exponent * | $m$ | - | 0.5 |
| Slope scaling exponent * | $n$ | - | 1 |
| Sediment erodibility * | $K_s$ | $1/m$ | 0.0002 |
| Bedrock erodibility * | $K_r$ | $1/m$ | 0.0001 |

*\* Parameters for SpaceLargeScaleEroder. See Shobe et al. (2017) for details.*

We can look at the fraction of material that comes from the northern region by analyzing the concentration value in channel bed sediment at four different locations marked in Figure 6b: the outlet of the entire catchment (black star), the outlet of a

southern sub-catchment (black diamond), the outlet of a middle sub-catchment (grey diamond), and the outlet of northern sub-catchment (white diamond). The southern sub-catchment only has a small portion of its headwaters in the red bedrock region, the northern sub-catchment is entirely within the red bedrock region, and the middle catchment has about 60% of its drainage area within the red bedrock region. Fluvial incision erodes red bedrock from the northern region. It is then transported downstream and deposited along the riverbed or removed from the domain entirely. After a period of transience, the sediment

colours within the catchment reach a steady state (Figure 6c). At this point, the "*sediment_property__concentration*" value reflects the fraction of channel bed material sourced from the northern region, shown in Table 6.

**Table 6. Concentration values at specific outlet points.**

| Outlet | % of catchment in red region | Bed sediment concentration value | Bedrock concentration value |
|---|---|---|---|
| Main channel | 33.33% | 0.3333 | 0 |
| Southern sub-catchment | 1.29% | 0.0129 | 0 |
| Middle sub-catchment | 60.43% | 0.6043 | 0 |
| Northern sub-catchment | 100% | 0.9999 | 1 |







**Figure 6. Example of fluvial sediment erosion, transport, and deposition changing the colour of channel bed sediment in a 2-dimensional erosional catchment. (a) A plan view hillshade overlain by a colour gradient showing topographic elevation. The shape of the river network is clearly visible. (b) A hillshade overlain by the bedrock colour. The northern region has a bedrock concentration value of 1 (coloured red) and the larger southern region has a value of 0, corresponding to a yellow colour. The four coloured streams are the main channels of the four sampled watersheds. The blue colour of the stream changes from light to dark with increasing drainage area. (c) A hillshade overlain by the surface sediment colour at steady state. Transport and deposition of red sediment from the north causes a reddening of channel bed sediment that decreases downstream as it is mixed with more and more sediment from the southern region. In all three maps, the outlet of the entire catchment is marked with a black star and each sub-catchment is delineated and has its outlet marked with a diamond (black: south, grey: middle, white: north).**

## 6 Potential applications

The *ConcentrationTracker* components allow the user to define the property of interest. Although the model is framed as a

mass balance, the "mass concentration" is unit agnostic and can also act as a volume concentration (e.g., volume of quartz





grains per volume of sediment) or a number concentration (e.g., number of atoms per volume of sediment). The colour concentration examples described above to illustrate the behaviour of the model can be changed to serve a wide variety of purposes.

However, since they depend on sediment fluxes calculated by other Landlab components, the concentrations must be properties that are physically transported as passive tracers, either as a fundamental feature of the sediment itself or as something physically sorbed to the sediment. Fluid tracers or chemicals transported in fluid cannot be simulated with the components presented here, though the same mass balance approach could be applied to a fluid flux component to achieve this. Chemical weathering of passive sediment tracers, however, can be handled in the bedrock weathering process and/or as user-defined

sources/sinks outside of the components themselves. As well, these components rely on the conceptual model of a landscape made up of a bedrock base overlain by a single homogeneous mobile regolith layer. Homogeneity requires an assumption of perfect mixing, which means that there can be no vertical variability in material or concentration values in the mobile layer. The fluxes are also comprised of homogeneous material, so there can be no differential mobility, either of sediments or of the properties assigned to the sediments. There is no ability for the property to move at any rate other than that of the bulk sediment

flux. With these constraints in mind, as long as the process components are well suited to the questions being asked, the *ConcentrationTrackers* can be used in many scenarios either specific to a single geomorphic process or when coupled together to simulate landscapes undergoing multiple geomorphic processes.

The *ConcentrationTracker* components were originally developed to study magnetic susceptibility in deposits sourced from regolith compared to those sourced from bedrock by tracking magnetite mass concentrations. Other provenance-style analyses

could measure detrital zircon, or any other mineral of interest. Different concentration fields can be applied to zircon counts, ages, or masses from one or many source populations. Alternatively, different concentration fields could be used to track the mass of different minerals across the same landscape. Similarly, the components can be used for movement and deposition of placer deposits and some specific types of soil contamination from known source areas. The latter is limited to contaminants sorbed to grains, as fluid contamination cannot be modeled.

One could also use *ConcentrationTracker* to model the luminescence characteristics of sediment, in which case the quantity of interest could be represented in terms of the "equivalent dose" of absorbed radiant energy per unit sediment mass required to reproduce an observed luminescence signal. For such an application, one would need to implement calculation of the gain of signal due to background ionizing radiation, and for the loss of signal due to bleaching by sunlight exposure (for an overview and 1D applications of this concept, see Gray et al., 2017, 2018, 2019).

Cosmogenic nuclide concentrations can be calculated by adding a source/sink term into the model loop to calculate production and decay rates. Multiple radioactive isotopes (e.g., $^{10}$Be-$^{26}$Al, Uranium-series) can be modeled by tracking multiple concentration fields and applying separate production/decay equations to each one. Examples of such applications using similar models can be found in Mudd (2017), Carretier et al. (2023), Petit et al. (2023), and Reed et al. (2023).

Although not a mass, the volume-averaged bulk age of sediment can be tracked as a number concentration within the mobile

sediment layer (e.g., Brosens et al., 2020). From a given starting time, all sediment and bedrock can be provided with ages that



increase through time. This property is transported with the sediment and averaged amongst mixing sediments, resulting in a volume-weighted average age for the sediment. This example is like our colour concentration examples. The *ConcentrationTrackers* apply to any property can be tracked by volume of grains, if variation of that property does not impact the parameters in the process components.

The erosion-deposition formulation of *SpaceLargeScaleEroder* allows modeling of alluvial deposits. Although concentration values become perfectly mixed within the deposit, a synthetic stratigraphy of sorts can be rebuilt by saving deposition rates and their related concentrations prior to mixing at each timestep.

In all cases, the effects of transient landscape response can be modeled.

Landlab is open source, and anyone can build a *ConcentrationTracker* component as long the companion sediment flux process
component is mass-conservative and fluxes can be tracked between grid nodes or on grid links.

## 7 Conclusions

We present a set of new numerical models to calculate passive sediment tracer concentrations in Landlab. These *ConcentrationTracker* components use a common mass balance foundation that is then adapted to couple with specific pre-existing sediment flux components in Landlab. This paper presents the *ConcentrationTrackerForDiffusion*, a companion
component to the *DepthDependentDiffuser* or *DepthDependentTaylorDiffuser* (used for linear and non-linear hillslope sediment transport, respectively) and the *ConcentrationTrackerForSpace*, a companion component to *SpaceLargeScaleEroder* (used for fluvial incision, transport, and deposition). The components can be coupled for use cases in which a multi-process landscape is desired.

The properties being tracked must be passive tracers of sediment physically transported with the sediment itself. All sediment
is assumed to always be homogeneously mixed. The components have numerous potential applications, such as calculation of erosion rates using cosmogenic radionuclide concentrations, provenance tracking using zircon counts, and sediment residence time calculations. We provide 1-dimensional and 2-dimensional examples of the *ConcentrationTrackers* for hillslope and fluvial domains that show how tracer concentrations evolve differently through time depending on the sediment transport process at play. The code for the examples is shown step-by-step in two accompanying Jupyter notebook user manuals.




## Appendix A

Table A1. List of variables.

| Variable | Units | Variable description |
| --- | --- | --- |
| $a$ | $L^2$ | Cell area |
| $B$ | L | Stream channel width |
| $C_X$ | M/ $L^3$ | Mass concentration ($C$) of property of interest ($X$) per volume of material denoted by subscript ($r$ : bedrock, $s$ : sediment, $sw$ : sediment entrained in water column, $p$ : sediment produced by weathering) |
| $D$ | $L^2$/T | Diffusivity constant |
| $D_{sw}$ | L/T | Rate of deposition of sediment from the water column, normalized by cell area |
| $E_r$ | L/T | Rate of erosion of bedrock, normalized by cell area |
| $E_s$ | L/T | Rate of erosion of sediment from the channel bed, normalized by cell area |
| $F_f$ | - | Fraction of fine sediment (becomes permanently suspended in water column) |
| $H_s$ | L | Depth of regolith layer |
| $H_d$ | L | Depth–decay constant for regolith production |
| $H^*$ | L | Regolith transport decay depth |
| $h_w$ | L | Depth of the water column |
| $K_r$ | $m^{-1}$ * | Bedrock erodibility |
| $K_s$ | $m^{-1}$ * | Sediment erodibility |
| $l$ | L | Streamwise length |
| $m$ | - | Area scaling exponent |
| $m_X$ | M | Mass ($m$) of property of interest ($X$) |
| $M_{X_{in/out}}$ | M/T | Rate of mass transfer ($M$) of property of interest ($X$) in/out |
| $n$ | - | Slope scaling exponent |
| $ncols$ | - | Number of columns in the gridded numerical domain |
| $nrows$ | - | Number of rows in the gridded numerical domain |
| $P_s$ | L/T | Soil production (bedrock weathering) rate |
| $P_0$ | L/T | Maximum soil production rate |
| $q_s$ | $L^2$/T | Sediment flux (normalized by width) |
| $Q_s$ | $L^3$/T | Sediment flux |
| $Q_{sw}$ | $L^3$/T | Sediment flux (sediment carried in water) |
| $Q_w$ | $L^3$/T | Water flux |
| $R$ | L | Bedrock elevation |
| $S$ | L/L | Local slope |
| $S_c$ | L/L | Critical slope |
| $t$ | T | Time |
| $U$ | L/T | Rock uplift rate |
| $V$ | L/T | Net effective settling velocity parameter |
| $V_s$ | $M^3$ | Volume of sediment |
| $\Delta x, \Delta y$ | L | x- and y- dimensions of a cell |
| $\eta$ | L | Topographic surface elevation |
| $\varphi$ | - | Porosity |
| $\Psi_X$ | M/T | Rate of mass gain or loss from sources/sinks |



**Code availability**

The source code for the current version of Landlab, including the *ConcentrationTrackerForDiffusion* and *ConcentrationTrackerForSpace* components, is available from the project website http://github.com/landlab/landlab (last access: June 2025) under the MIT License. Landlab's documentation, including installation instructions and software dependencies can be found at: https://landlab.csdms.io/ (last access: June 2025). The static version of Landlab used to produce the results in this paper are archived on Zenodo under https://doi.org/10.5281/zenodo.15652866 (Roberge, 2025b). The scripts to run the components and produce the plots for all the simulations presented in this paper are archived on Zenodo under

https://doi.org/10.5281/zenodo.15653060 (Roberge, 2025a).

**Interactive computing environment**

Two Jupyter notebooks serve as user manuals. They describe how to use the model components and show step-by-step instructions and code that walk through simplified versions of the 1D and 2D example applications presented in this paper. The simplified examples are adapted to run more quickly, so use less physically realistic parameter values, but show the same

general results. They can be found at: https://github.com/loroberge/pub_Roberge_et_al_ConcentrationTracker_GMD/.

**Author contribution**

LR, NG, BC and GT conceptualized the model. LR developed the Landlab model code with help from BC. LR designed and performed the simulations, wrote the original draft, and prepared the manuscript with contributions from all co-authors.

**Competing interests**

The authors declare that they have no conflict of interest.

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
