# Peer review of "ConcentrationTracker: Landlab components for tracking material concentrations in sediment"

_EGUsphere, 2025_

## Author Response (AR1)

Dear Sebastien Carretier,

We thank you for your positive feedback and shared view of the motivation for building our model. Your insightful review comments are greatly appreciated.

We have attached a PDF containing our responses. The comments are quoted in "*black italics*", and our answers are in blue.

Sincerely,

Laurent Roberge

**Response to "RC1: 'Comment on egusphere-2025-2445', Sebastien Carretier, 26 Oct 2025":**

*"I think there is a simple experiment missing that would demonstrate the validity of the model, which I proposed in my 2016 paper: placing tracers on a pixel at the top of an inclined plane with a constant slope and only diffusion (and no uplift). In this case, we have a simple analytical solution that links the evolution of the spatial standard deviation of the concentration with the diffusion coefficient and time (Einstein's formula). By comparing the theoretical predictions with the Landlab results, you could show that the model gives consistent results, and perhaps independent of the model's time step and space step. It would be useful to discuss the dependence of the new module on these two parameters."*

We agree with the premise of the simple experiment, however the implementation of our model is sufficiently different to that of Carretier et al., (2016) that we believe such an experiment would not achieve the desired validation. The ConcentrationTrackers are essentially mixing models in which the tracer concentrations exist as a continuum field linked to the sediment flux field. My understanding of Cidre is that the probability of clast movement is calculated separately from sediment flux as a function of erosion and deposition (hence the value of the validation test to show that the particles behave in the predicted manner). I believe such a test for the ConcentrationTrackers would be a validation of the diffusion equation itself (i.e., the DepthDependentDiffuser or DepthDependentTaylorDiffuser). The diffusion equations implemented in Landlab (to which the ConcentrationTracker are coupled) were developed, tested, and published prior to and independently of the ConcentrationTracker.

*"Line 27 «few LEMs account for the storage, fate, and transport of other sediment properties" : which one ?"*
& *"Line 29-30 I agree, this was my motivation for the grain tracers in Cidre in the 2016 paper."*

We have added a few references pointing the reader to the other models with similar motivation and designs.

*"Line 205 In the equation how $Cxpi^{t+1}$ is known ? and in the line bellow it is written that the "remaining unknown is $CXs^{t+1}$ on both sides of the equation" but I do not see it. Is there a tipo?"*

This was a typo left over from an intermediate step in a draft version of the model description. We have updated the text to describe $Cxpi^{t+1}$ and removed the typo.

*"Line 250. Could you explain just a bit more how to obtain this equation?"*

We have now added text pointing the reader to the original SPACE model paper (Shobe et al., 2017) where this local analytical equation is described in more detail.

Dear Anonymous Referee #2,

We thank you for your detailed feedback and review comments. In particular, your suggestions on improving the figures are greatly appreciated.

We have attached a PDF containing our responses. The comments are quoted in "*black italics*", and our answers are in blue.

Sincerely,

Laurent Roberge

**Response to "RC2: 'Comment on egusphere-2025-2445', Anonymous Referee #2, 29 Oct 2025":**

*Line 72–73: I would reformulate this as: "with a single-surface detachment-limited approach similar to..."*

We reformulate the sentence as suggested.

*Line 100: For "we take a more generalized approach," I suggest explicitly stating what you add:*
- *Any user-defined tracer can be used, rather than only very specific ones.*
- *Landslide deposits and other processes are also included via geomorphologic units in Landlab.*

*This helps the reader immediately see the added value of this tool.*

We change the paragraph as suggested to explicitly state what we mean by a "generalized approach" and how the ConcentrationTrackers fit in with Landlab's geomorphic process components.

*Lines 130–140: I think the article would benefit greatly from a figure explaining these concepts. Table 1 could be integrated into that figure (I made a quick sketch of what this could look like in the attachment).*

We point the reader to the Tucker & Slingerland (1997) paper that has a similar figure and explains in detail the reasoning behind this approach.

*Line 135: Use "neighbouring" instead of "neighboring." I have not checked the entire article, but there may be other inconsistencies between American and British spelling throughout.*

We have changed American spelling to British spelling throughout.

*Figure 1: I think a few additions could make this figure clearer (see attachment):*
- *The incoming and outgoing arrows should be horizontal and enter/exit at the centers of the box sides.*
- *Add dotted-line boxes to illustrate the 2D grid.*
- *Indicate the area "a."*
- *Perhaps also indicate the weathering process visually.*

Thank you for the great figure suggestions (here and for figures 2, 3, 4, and 5). We have updated Figure 1 taking these points into account.

*Figure 2: Similarly, some minor adjustments could greatly improve readability (see attached suggestion).*

We have updated Figure 2 taking these points into account.

*Line 243: I suggest rewriting this as: "Over large timescales, the relative change in sediment concentration for a given water column becomes negligible. This implies that our concentration trackers can only be used over large timescales."*

*It is important for future users to understand the timescales for which these concentration trackers are valid.*

We have adjusted the wording to reflect this.

*Line 307: It would be useful to provide estimates of the timescales over which the assumption of homogeneous mixing holds, so that future users know when the concentration trackers can and cannot be applied. I suggest citing sources about homogenous mixing to back up this statement.*

We have added relevant timescales and citations.

*Figure 3: I have several suggestions:*
- *Rename the examples from "Example 1," "2," "3" to "High diffusivity," "Steep hillslope," and "High soil production ratio" to make the figure's message clearer.*
- *Draw boxes around A&D, B&E, and C&F, with the titles or example names above the boxes.*
- *Panel G:*
- *Change the y-axis label to "Tracer concentration at x=0."*
- *Use a different color scheme (e.g., #D81B60, #DC267F, #FFB000) so it is not confused with the time color scheme from Fig. 3d–f.*
- *If possible, indicate the sediment input at the 150 m marker.*

We have updated Figure 3 taking these points into account.

*Figure 4: Apply similar changes as for Figure 3. You might rename the titles to "Slow settling speed" and "High settling speed."*

We have updated Figure 4 taking these points into account.

*Figure 5: I think panels A and B could be removed, as they do not add information essential for the reader (amusingly, the mountain shape looks a bit like a cat).*

Although we agree that these panels are not necessarily essential, we think they provide valuable context, especially for a reader who is unfamiliar with the processes involved in the scenario. The hillshade provides a sense of the verticality of the slope that is not clear in panels C and D. The picture in panel D is intended to visually show a real-world equivalent of the model scenario, which may not be obvious to the reader solely from a description in words.

*Figure 6:*
- *Consider merging the panels into two plots:*
- *Bedrock regions with drainage area. Legend: diamond = subcatchment outlet, star = main river outlet, dotted line = subcatchment boundary, color scale from white to dark blue representing drainage area, colour scale of bedrock colour.*
- *Elevation model with hillshade overlay and river surface colors. Currently, the color distribution in different rivers is not visible in the north. Legend: elevation color scale and fraction of red color scale.*

We have updated Figure 6 taking these points into account.

*Line 452: You could also add that multiple tracers can be tracked simultaneously.*

We now state this at point later in the section.